# Study of the psychometric properties of the HLS-EU-12 questionnaire in rural Bangladesh

**Fakir M. Amirul Islam** [1,2]*

1 School of Health Sciences, Swinburne University of Technology, Hawthorn Victoria, Australia,
2 Organisation for Rural Community Development (ORCD), Dariapur, Narail, Bangladesh

* fislam@swin.edu.au

## Abstract

### Objectives

Poorer health literacy leads to limited knowledge of diseases and lower adherence to medication. Several tools are available for measuring health literacy, including the HLS-EU-Q12 questionnaire. This study aimed to examine the psychometric properties of the HLS-EU-Q12 questionnaire using Rasch measurement theory to assess general health literacy among adults in rural Bangladesh.

### Materials and Methods

Baseline data was collected through a cluster randomized control trial (RCT) in the Narail district of Bangladesh from 1 December 2020 to 31 January 2021, involving 307 adults aged 30–75 with hypertension. The HLS-EU-Q12 questionnaire, which has 12 items, was validated using the Rasch measurement model and analyzed using RUMM2030. The analysis was focused on differential item functioning (DIF) across gender and age, targeting, multidimensionality, response dependency, and item categorization.

### Results

The sample consisted of almost equal proportions of females (50.2%) and males (49.8%), and control (50.8%) and intervention participants (49.2%). Initial examination indicated that the tool had a poor overall fit with the Rasch model, shown by a significant item-trait interaction ($\chi2 = 100.5$ df = 48, p < 0.001). The reliability, measured by person separation index (PSI), was 0.746, which was considered satisfactory. The overall item fit residual (IFR) (M = 0.236, SD = 1.318) and the person fit residuals (PFR) (M = -0.186, SD = 1.03) were within the acceptable range of SD ± 1.4. All items were found to have ordered thresholds, suggesting that respondents had no difficulty differentiating between response options on the 4-point Likert-type scale. Only item 12 ("Join a sports club or exercise class if you want to") had a fit residual

**Data availability statement:** The supporting information file is in the text before reference. Data S1 HLS EU_PLOS One.xlsx

**Funding:** The author(s) received no specific funding for this work.

value outside the acceptable range. Removing item 12 resulted in a good overall fit ($\chi2 = 60.35$ df = 44, p = 0.05) and a slight improvement to the PSI (0.746 for 12 items vs.0.756 for 11 items). None of the items showed significant DIF for age and gender. Only two items showed residual correlation coefficient 0.20, indicating possible redundancy. The analysis demonstrated the scale's unidimensionality, as shown by the lower bound of a binomial 95% confidence interval (CI) for the observed proportion (5.2%) being within the 95% confidence limit.

## Conclusion

The study demonstrated the potential effectiveness of the HLS-EU-Q12 as a tool for assessing health literacy among adults in Bangladesh. However, further study is needed to evaluate the tool across different populations, including an in-depth investigation of item 12 to determine its inclusion or removal.

## Trial registration

ClinicalTrials.gov NCT04505150. Registered on 7 August 2020.

## Introduction

Health literacy is critical for improving health and well-being and reducing health inequities [1,2]. In recent years, researchers have paid increasing attention to health literacy, and it was listed as the core topic at the 9th Global Conference on Health Promotion in 2017 [3,4]. The prevalence of health literacy has been widely reported, but findings remain inconsistent. In a systematic review reporting the global prevalence of health literacy among adult diabetes patients, Abdulla et al. [5,6] reported the prevalence of limited health literacy between 7.3% and 82%. One of the main reasons for such variation in prevalence is the use of different measurement tools.

Several tools for measuring health literacy exist, including general and disease-specific health literacy instruments [7,8]. In a systematic review summarizing health literacy instruments, Tavousi et al. reported 39 general, 90 condition-specific (disease or content), 22 population-specific, and 11 electronic health literacy instruments [9]. However, not all tools were developed using modern guidelines, such as item response theory and Rasch modeling, and there is no 'gold standard' measure of health literacy [10,11]. Examples of the health literacy tools available include the Brief Health Literacy Survey BHLS [12] [four items], which is a self-report instrument developed to identify patients with insufficient health literacy; the Short Test of Functional Health Literacy in Adults (s-TOFHLA) [13] [36 items], which tests a patient's ability to read using reading materials from the health care setting, the Rapid Estimate of Adult Literacy in Medicine (REALM) [14] [66 items], for quickly screening proficiency in reading health terms, and the eHealth Literacy Scale (eHEALS) [15] [8 items], which evaluates consumers' perceived skills in using information technology for health. Despite the availability of these tools, there is no comprehensive

tool with a reasonable number of items to measure health literacy for a common purpose. Health literacy measurement should be standardized across countries, to develop adequate country-specific measures and support benchmarking [16]. Moreover, studies suggest that well-constructed short scales can be more robust predictors than lengthier instruments or interviews[17,18].

Among the available tools, the most widely used tool is the European Health Literacy Questionnaire (HLS-EUQ) developed by the HLS-EU Consortium [8]. The HLS-EU-Q and its various versions (HLS-EU-Q86/HLS-EU-Q47/ HLS-EU-Q16/ HLS-EU-Q6) [8] are based on a comprehensive definition of health literacy and measure it across different populations [19]. Finbråten et al. established the HLS-EU-Q12 short version from the HLS-EU-47 by applying Rasch modeling and confirmatory factor analysis, which summarized the four cognitive domains from the HLS-EU-47 and proposed a feasible health literacy screening tool[20]. Duong et al. developed a 12-item short-form health literacy questionnaire (HLS-SF12) from the HLS-EU-Q47, retaining the conceptual framework that accounted for 90% or more of the variance of the complete form in each country [21]. Using the HLS-EU questionnaire, several studies have been conducted in Asia [22–25]. However, none of the studies conducted in Asia have used the Rasch analysis to validate the tools. Nguyen et al. have provided evidence to support using Rasch analysis in health literacy instrument development [10]. It is necessary to validate a questionnaire before its application to a particular population.

Bangladesh is densely populated, with approximately 163 million people, including 27 million adults with hypertension and 6–8 million (~9%) with diabetes [21,22]. The literacy rate of people who can read, write, and understand a short, simple statement about their daily lives among adults aged 15 years or older was 74.9% in 2019 [23]. Previous studies have reported that rural people did not have sufficient knowledge of the most common chronic diseases, such as diabetes, common eye diseases, and mental health [24–26]. Research on health literacy in Bangladesh is minimal. Health service delivery is complex, involving a mix of public, private, and informally trained healthcare providers. Additionally, Bangladesh's education system is not equipped with the necessary skills to help people effectively interact with modern health systems and information to improve their health [27]. Studies suggest that specific interventions can improve health literacy [28]. According to the author, in Bangladesh, only Salway et al. studied health literacy among non-communicable disease service seekers using the HLS-EU-Q16[29,30]. However, the HLS-EU-Q16, a 16-item tool, was found to have issues related to item misfit, DIF, and unordered response categories[30]  . In contrast, the HLS-EU-Q12 demonstrates stronger psychometric properties and was tested in six Asian countries using Rasch analysis [20,31].

A brief and validated tool is needed to assess health literacy and conduct appropriate interventions [10]. The current research aims to validate the brief health literacy questionnaire HLS-EU-Q12 in a rural area of Bangladesh by using Rasch analysis.

## Materials and Methods

### Research Design

A cluster randomized control trial was conducted to assess the effectiveness of lifestyle changes in managing blood pressure. A total of 307 participants, 156 in the control group and 151 in the intervention group, aged 30–75, were recruited at baseline between 1 December 2020 to 31 January 2021. The current study is based on the baseline data only, therefore is not expected to have any intervention effects [32].

### Study subjects

To conduct the cluster RCT, participants were selected from the first cross-sectional Bangladesh Population-based Diabetes and Eye Study conducted in 2013, details of which were presented previously [33]. To provide context on the study location, Bangladesh has 64 districts. Each district is divided into 3–7 Upazilas, which are further divided into

10–15 Unions, and each Union consists of 15–20 villages [34]. Participants who had blood pressure greater or equal to 130/80 mm Hg at the time of screening were eligible for the current study [35].

The inclusion criteria were:

1. Clinic blood pressure ≥130/80 mm Hg in individuals not taking medication.

2. Blood pressure <130/80 mm Hg but using anti-hypertensive medicines for at least six weeks.

3. Living in the Banshgram Union only.

The exclusion criteria were:

1. Aged over 75 years.

2. Pregnant.

3. Advanced cardiovascular diseases (CVDs) or any severe condition that restricted participation in the study.

These exclusions were applied because older individuals, pregnant women, and individuals with advanced CVDs might face challenges participating in the lifestyle modification program.

## Sample size and statistical power

The sample consisted of 307 adults recruited for a randomized cluster trial aimed at managing blood pressure by changing lifestyle in the Narail district. After recruitment, participants were randomly assigned, with 156 in the control arm and 151 in the intervention arm. Participants from the control group could be neighbors of those in the intervention group, which would lead to contamination between groups. As a result, we opted to implement a cluster randomized controlled trial (RCT) instead of individual randomization. The sample size was calculated to detect a minimum difference of 3 mm Hg in blood pressure, assuming a standard deviation of 6 mm Hg [36]. The current research is based on the fixed sample size for the intervention study. A sample size of approximately 300 is recommended for a Rasch analysis, as larger sample sizes may lead to type 1 errors, which can incorrectly reject an item for not fitting the Rasch model [37]. A sample size 300 provides a 99% confidence level that the estimated item difficulty will be within ±½ logit of its stable value [38]. Based on the assumption that the participants could have a health literacy level of 50%, the sample was sufficiently large enough to detect as minimum difference of 8%, an arbitrary threshold, in the proportion of attaining health literacy between males and females, and between those with no schooling and primary or secondary level of education (statistical power > 80%, p = 0.05). The formula for calculating the sample size was $n = p^* \cdot q^* (z_{\alpha/2}/E)^2$, where p = 0.50, q = 0.5, $z_{\alpha/}$ = 1.96 (for 95% confidence interval), E = 0.08. Given these parameters, the required sample size was 150 for males or females. Therefore, the total sample size of 307 was adequate for the current research to fulfill both the objectives of the psychometric validation of the tool for measuring health literacy.

## Recruitment and Data Collection

A local NGO facilitated the recruitment process. The NGO investigators and trained data collectors communicated with potential participants via telephone or direct in-person contact. Data was collected at schools or community halls suitable for interviewing participants and collecting blood pressure data. The NGO investigators and four trained data collectors collected data from face-to-face interviews. The recruitment and the study protocol were published elsewhere [32]. Baseline data was collected through face-to-face interviews to minimize response bias and ensure that participants with limited education could understand the questionnaires. The study area was divided into two clusters to conduct the cluster RCT. The first cluster, consisting of nine villages and 151 participants, was designated as the intervention group. The control group was the second cluster, comprising the remaining nine villages and 156 participants.

## Questionnaire preparation

A local senior educator and the principal investigator, each translated the questionnaire into Bengali. For data collection, the two translated versions were combined and finalized with the questionnaire, by mutual agreement between the two translators. No formal text analysis was conducted to assess the agreement between the two translators. However, a pilot test was carried out among ten individuals to evaluate the questionnaire's comprehension, wording, and appropriateness. This group included five females, and five individuals aged 50 years or older with hypertension who were not involved in this study. The pilot test did not require any significant adjustments.

## Health Literacy Survey Tool: HLS-EU-Q12

The study questionnaire comprised several sections [32], including socio-demographic factors and the short form of the European Health Literacy Survey questionnaire (HLS-EU-Q12) [20]. Sørensen et al. [8] developed the HLS-EU-Q47 tool with 47 items with a 4-point rating scale with response categories ranging from easy (rating point 1) to challenging (rating point 4) to assess health literacy among the general population. Finbraten et al. validated the HLS-EU-Q47 version [20] and proposed the 12-item short version (HLS-EU-Q12), in which the rating scales were reversed. This consisted of response categories ranging from very difficult (rating point 1) to very easy (rating point 4). The questions and responses are shown in Table 1 in the results section.

## Ethics approval

The institution's Human Research Ethics Committee approved the study protocol. The participants were provided with written information about the project and had full rights to withdraw from the study at any stage. The investigators obtained written consent from all participants.

## Rasch Measurement Theory

The first outcome of this study was the validation of the HLS-EU-Q12 tool. The Rasch analysis in this study was conducted using the RUMM 2030 (Rasch Unidimensional Measurement Models, Perth, WA, Australia) package[39]. The partial credit parameterization was used for data analysis. The partial credit model (PCM) is a generalisation of the dichotomous Rasch model that recognizes that each item has its own rating scale structure. For all analyses the average item-location estimate was set to 0.0, and the standard deviation to 1.0. Rasch measurement theory applies several tests to examine the scale's psychometric properties, including the model's reliability, item fit, ordering of response categories, the presence of DIF, local independence, unidimensionality, and targeting[40]. Rasch analysis provides person locations, item difficulties, and thresholds for each item[41]. When applying Rasch analysis with polytomous items, the first stage is assessing whether the thresholds for all items are ordered. Disordered thresholds usually occur when individuals find it challenging to distinguish [42,43].

**Model fit.** The model's overall fit is described by the Chi-square item-trait interaction statistics, which are ideally non-significant [44]. A non-significant p-value indicates that the items' hierarchical orders are stable across all underlying attribute levels. Item-person interaction statistics are a standardized normal distribution with a mean of zero and SD of 1, indicating a perfect model. If the overall model has SD values of 1.5 or more for either item or person, it suggests potential issues with model fit. Individual item and person fit statistics are presented as residuals based on comparisons between observed and expected values. Residual values within ±2.5 SD are considered acceptable, while fit residual values beyond ±2.5 SD indicate item misfit or overfit.

**Reliability.** The Person Separation Index (PSI) [45] was used to check the reliability and internal consistency of the model. The PSI is analogous to Cronbach's α but is based on a non-linear transformation of the raw scores and can be measured in the presence of missing values. A high PSI value indicates high reliability, suggesting that the scale can

**Table 1. Responses to HLQ-EU questionnaire by the total participants and by study types, control vs. intervention group.**

| HLQ-EU question | Response | Total, N=307 | | Control, N=156 | | Intervention, N=151 | | |
|---|---|---|---|---|---|---|---|---|
| | | N | % | n | % | n | % | P* |
| "…find information on treatments of illnesses that concern you?" | Very difficult | 9 | 2.9 | 6 | 3.8 | 3 | 2.0 | |
| | Difficult | 98 | 31.9 | 53 | 34.0 | 45 | 29.8 | 0.39 |
| | Fairly easy | 191 | 62.2 | 91 | 58.3 | 100 | 66.2 | |
| | Easy | 9 | 2.9 | 6 | 3.8 | 3 | 2.0 | |
| "…understand the leaflets that come with your medicine?" | Very difficult | 57 | 18.6 | 32 | 20.5 | 25 | 16.6 | |
| | Difficult | 134 | 43.6 | 68 | 43.6 | 66 | 43.7 | 0.72 |
| | Fairly easy | 85 | 27.7 | 37 | 23.7 | 48 | 31.8 | |
| | Easy | 31 | 10.1 | 19 | 12.2 | 12 | 7.9 | |
| "…judge the advantages and disadvantages of different treatment options?" | Very difficult | 12 | 3.9 | 7 | 4.5 | 5 | 3.3 | |
| | Difficult | 115 | 37.5 | 59 | 37.8 | 56 | 37.1 | 0.24 |
| | Fairly easy | 170 | 55.4 | 89 | 57.1 | 81 | 53.6 | |
| | Easy | 10 | 3.3 | 1 | 0.6 | 9 | 6.0 | |
| "…call an ambulance in an emergency?" | Very difficult | 22 | 7.2 | 16 | 10.3 | 6 | 4.0 | |
| | Difficult | 105 | 34.2 | 59 | 37.8 | 46 | 30.5 | 0.03 |
| | Fairly easy | 126 | 41.0 | 60 | 38.5 | 66 | 43.7 | |
| | Easy | 54 | 17.6 | 21 | 13.5 | 33 | 21.9 | |
| "…find information on how to manage mental health problems like stress or depression?" | Very difficult | 14 | 4.6 | 4 | 2.6 | 10 | 6.6 | |
| | Difficult | 132 | 43.0 | 78 | 50.0 | 54 | 35.8 | 0.44 |
| | Fairly easy | 152 | 49.5 | 69 | 44.2 | 83 | 55.0 | |
| | Easy | 9 | 2.9 | 5 | 3.2 | 4 | 2.6 | |
| "…understand why you need health screenings (such as breast exam, blood sugar test, blood pressure | Very difficult | 29 | 9.4 | 14 | 9.0 | 15 | 9.9 | |
| | Difficult | 133 | 43.3 | 67 | 42.9 | 66 | 43.7 | 0.62 |
| | Fairly easy | 139 | 45.3 | 71 | 45.5 | 68 | 45.0 | |
| | Easy | 6 | 2.0 | 4 | 2.6 | 2 | 1.3 | |
| "…judge which vaccinations you may need?" | Very difficult | 73 | 23.8 | 34 | 21.8 | 39 | 25.8 | |
| | Difficult | 186 | 60.6 | 99 | 63.5 | 87 | 57.6 | 0.70 |
| | Fairly easy | 45 | 14.7 | 21 | 13.5 | 24 | 15.9 | |
| | Easy | 3 | 1.0 | 2 | 1.3 | 1 | 0.7 | |
| "…decide how you can protect yourself from illness based on advice from family and friends?" | Very difficult | 1 | 0.3 | 0 | 0.0 | 1 | 0.7 | |
| | Difficult | 27 | 8.8 | 10 | 6.4 | 17 | 11.3 | 0.07 |
| | Fairly easy | 203 | 66.1 | 103 | 66.0 | 100 | 66.2 | |
| | Easy | 76 | 24.8 | 43 | 27.6 | 33 | 21.9 | |
| ) "…find out about activities (such as meditation, exercise, walking, Pilates, etc) that are good for your mental well-being?" | Very difficult | 10 | 3.3 | 6 | 3.8 | 4 | 2.6 | |
| | Difficult | 108 | 35.2 | 61 | 39.1 | 47 | 31.1 | 0.16 |
| | Fairly easy | 176 | 57.3 | 82 | 52.6 | 94 | 62.3 | |
| | Easy | 13 | 4.2 | 7 | 4.5 | 6 | 4.0 | |
| "…understand information in the media (such as Internet, newspaper, magazines) on how to get healthier?" | Very difficult | 36 | 11.7 | 22 | 14.1 | 14 | 9.3 | |
| | Difficult | 143 | 46.6 | 69 | 44.2 | 74 | 49.0 | 0.71 |
| | Fairly easy | 107 | 34.9 | 48 | 30.8 | 59 | 39.1 | |
| | Easy | 21 | 6.8 | 17 | 10.9 | 4 | 2.6 | |
| "…judge which everyday behavior (such as drinking and eating habits, exercise, etc.) is related to your health?" | Very difficult | 10 | 3.3 | 3 | 1.9 | 7 | 4.6 | |
| | Difficult | 100 | 32.6 | 60 | 38.5 | 40 | 26.5 | 0.43 |
| | Fairly easy | 189 | 61.6 | 88 | 56.4 | 101 | 66.9 | |
| | Easy | 8 | 2.6 | 5 | 3.2 | 3 | 2.0 | |

*(Continued)*

**Table 1.** (Continued)

| | | Total, N=307 | | Control, N=156 | | Intervention, N=151 | | |
|---|---|---|---|---|---|---|---|---|
| "… join a sports club or exercise class if you want to?". | Very difficult | 12 | 3.9 | 5 | 3.2 | 7 | 4.6 | |
| | Difficult | 133 | 43.3 | 66 | 42.3 | 67 | 44.4 | 0.37 |
| | Fairly easy | 156 | 50.8 | 81 | 51.9 | 75 | 49.7 | |
| | Easy | 6 | 2.0 | 4 | 2.6 | 2 | 1.3 | |

*Boneferroni adjusted p= 0.004 (0.05/12).

distinguish individuals along the latent trait [46]. A PSI value above 0.75 indicates good reliability, while values above 0.9 indicate very good or excellent reliability [47].

**Unidimensionality.** One of the assumptions of Rasch analysis is unidimensionality, which occurs when all items measure only one common attribute only [48]. Three steps are involved to test unidimensionality within the Rasch model:

1. Identify two item sets from a Principal Component Analysis (PCA) of residuals

2. Estimate separate person measures based on the two item sets

3. Compare the two estimates on a person-by-person basis using *t*-tests and determine the number of cases that differ significantly at the 0.05-level.

If ≤5% of tests are significant, or the lower bound of a binomial 95% confidence interval (CI) of the observed proportion overlaps 5%, then the scale is considered unidimensional. Otherwise, the scale is regarded as multidimensional [49].

**Local Independence.** The local independence (LD) can be tested by calculating the correlation coefficients among residuals, which are expected to be unrelated and generally close to zero [50]. Violations of LD can lead to inflated estimates of reliability and problems with construct validity. Currently, no well-documented suggestions of a single critical value indicate LD. Marais and Andrich[51] investigated dependence at a critical value of 0.1, but a value of 0.3 has often been used [52–54]; critical values of 0.5 [55,56] and even 0.7 [57] were also found to be in use. Christensen et al. [58] conducted a comprehensive study to identify LD in the Rasch model using residual correlations and reported that LD depended on several factors, including the number of items, number of response categories, and the number of respondents. For example, for polytomous items, the critical value for LD was approximately 0.11 for 10 items, 0.18 for 15 items, and 0.21 for 20 items and 300 participants for each item condition. The study reported a minor effect on the number of items, with a decrease in critical values with an increase in sample size.

**Differential Item Functioning (DIF).** In Rasch's measurement theory, a scale should work equivalently across different groups, irrespective of factors such as age or gender. Invariability indicates that 'items are not dependent on the distribution of persons' abilities and the person's abilities are not reliant on the test items. The items are expected to produce DIF if some factors do not display an equal likelihood of location estimate. Any presence of DIF violates the requirement of independence [59]. To examine DIF, a two-way analysis of variance (ANOVA) of standardized residuals is used to determine if there is a significant difference in mean residuals across levels of relevant person factors [60]. In the present study, participants were categorized as either adults (aged 30–59) or older adults (aged 60–75), and by gender (male or female) to test the DIF of the tool.

**Targeting.** In Rasch analysis, targeting is a key aspect of developing or validating a scale. It refers to how well the scale captures a person's estimates based on location. The item's difficulty and a person's ability are assessed by graphically and statistically comparing the distribution of the item threshold map. A scale is considered well-targeted if the mean person location values are around zero [61]. A positive value indicates better health literacy than is captured by the

average difficulty of the items. Conversely, poor targeting can lead to lower reliability and reduced capacity of the scale to distinguish between individuals based on their ability [46].

**Statistical analysis.**  Chi-square tests were used to detect any association between the study groups (control and intervention) and individual health literacy items. The statistical software SPSS (SPSS Inc, version 27) was used for the Chi-square tests, and RUMM2030 (2010) software was used for Rasch analysis.

## Results

### Participants' profile and health literacy in individual item

Of the 307 participants, 154 (50.2%) were female, 206 (67%) were below 60 years of age, and 156 (50.8%) were in the control cluster (data are not shown in a Tabular format). Health literacy in individual items for the total sample and by study groups are shown in Table 1. Approximately 40% of the participants reported difficulty with most of the items. The item "…judge which vaccinations you may need?" was the most difficult item (very difficult 23.8% and difficult 60.6%) and "…decide how you can protect yourself from illness based on advice from family and friends?" was the easiest item (fairly easy 66% and easy 24.8%). This baseline data showed no difference between the intervention and control groups.

### Psychometric properties of the HLS-EU-Q12

**Model fit.**  Initial inspection showed poor overall fit with the Rasch model, as indicated by a significant item-trait interaction ($\chi2 = 100.5$ df $= 48$, $p < 0.001$). However, the item fit residual (IFR) (M $= 0.236$, SD $= 1.318$) and the person fit residuals (PFR) (M $= -0.186$, SD $= 1.03$) were within the acceptable range of SD $\pm 1.4$ (Table 2). The value of the PSI (analogous to Cronbach's alpha) for all items with four response categories was 0.746, indicating that the scale demonstrated acceptable reliability in distinguishing between individuals. All 12 items were found to have ordered thresholds (Fig 1), suggesting the respondents have no difficulty differentiating between the response's choices with the 4-point Likert-type scale used in the HLS-EU-12 scale.

**Item fit.**  Individual item fit statistics showed that only item 12 ("join a sports club or exercise class if you want to") had fit residual values (mean $= 0.15$, SD $= 2.903$, $p < 0.001$) outside the acceptable range of $\pm 2.5$ standard deviations (Table 3). All other items fit the residual within the acceptable range. Further analysis by removing item 12 showed a good overall fit with the Rasch model, as indicated by an insignificant item-trait interaction ($\chi2 = 60.35$ df $= 44$, $p = 0.05$) The PSI value for the 11 items with the same four response categories was 0.756, which was a minor improvement from 0.746, indicates that removing item 12 had no substantial adverse effect on the scale (Table 2).

**Differential item functioning.**  Table 3 shows the differential item function for gender and age groups for all items and after deleting item 12. None of the items showed significant DIFs for age and gender with all items or after removing item 12, indicating the scale works equally for males and females and adults and older adults.

**Table 2. Overall model fit statistics of the HQL scale.**

| Model fit statistics. | N $= 306$, item $= 12$ | N $= 306$, item $= 11$ |
| --- | --- | --- |
| Overall model fit, Chi-square value | 100.5 | 60.35 |
| Degree of freedom (DF) | 48 | 44 |
| Bonferroni adjusted p value (0.004). | 0.00001 | 0.05 |
| Item fit residuals (mean (SD)) | 0.236 (1.318) | 0.232 (0.999) |
| Person fit residuals (mean (SD)) | -0.186 (1.03) | -0.221 (1.08) |
| Person separation index (PSI) | 0.746 | 0.756 |
| Unidimensionality test (% that goes beyond 95% CI) | 5.2% CI (2.8–7.7) | 5.2% CI (2.8–7.8) |

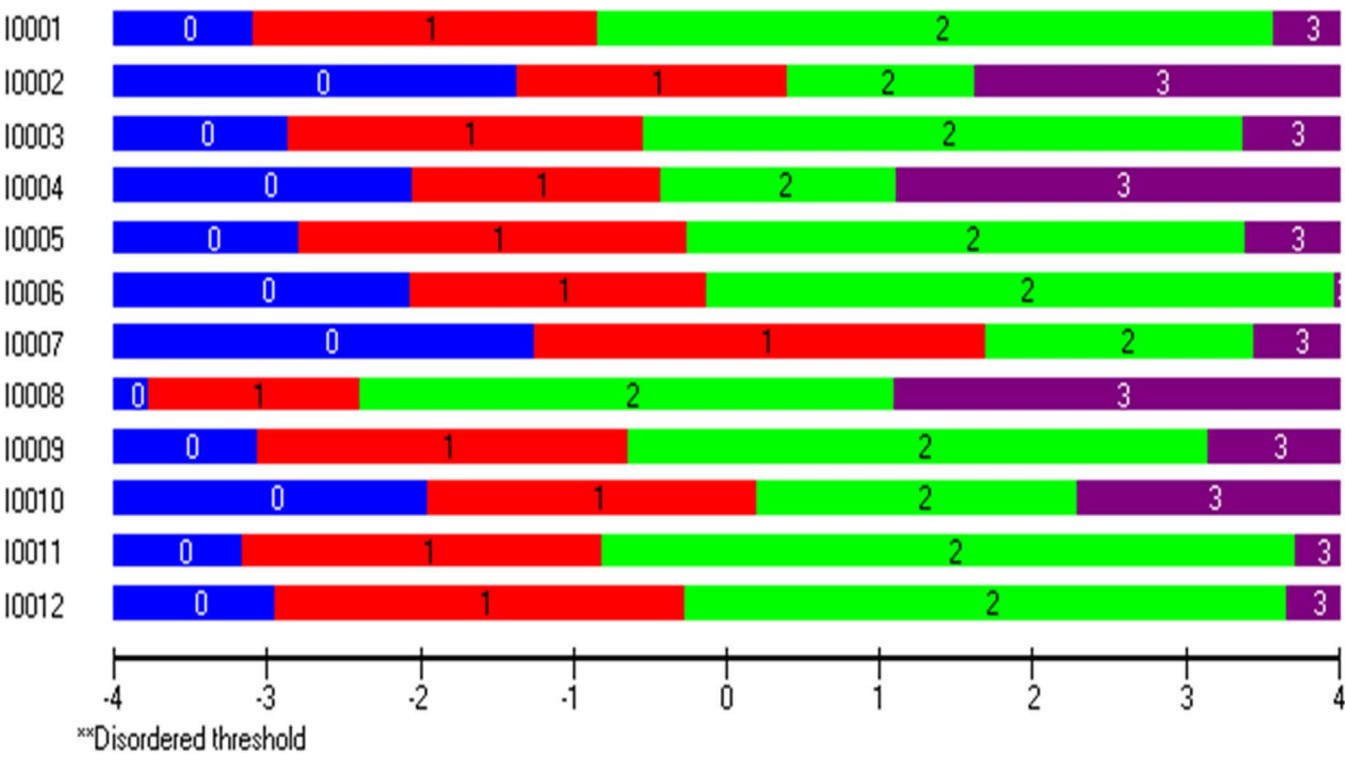

**Fig. 1. Threshold maps of the HLS-EU-Q12L scale validated in a rural district in Bangladesh.**

**Unidimensionality.** The Unidimensionality of HLS-EU-12 was tested using principal component analysis (PCA) residuals for total items (5.2%, 95% CI 2.8% to 7.7%) and 11 items, excluding item 12 "..join a sports club or exercise class if you want to.." (5.2%, 95% CI 2.8% to 7.8%) indicating the lower bound did not fall within the 5% of the critical region (Table 2 (last row)), which supports the Unidimensionality of the HLS-EU-12 item.

**Local Dependency.** Table 4 shows the residual correlation matrix. Of the 12x12 residual correlation matrix, all correlation coefficients lay between 0.02 to 0.10, except the correlation coefficient was 0.13 for item 1 with 3, item 2 with 10, and 0.20 for item 5 with 9. Sub-analysis by excluding item 5 and keeping item 9 or vice versa showed that none of the residual correlations were larger than 0.13, indicating items 5 and 9 had some redundancy information. Thus, all the items can be retained with caution.

**Targeting.** Fig. 2 shows the person-item distribution for all items and Fig. 3 for items 1–11, which displays the level of location of a person's ability and item difficulties, respectively. The targeting of persons and items was assessed by observing the person-item location distribution map by plotting the person and item locations on the same continuum and the item characteristic curve. The distribution is shown for females and males with different colors. The figures show that the item set, and the persons are well spread around the mean, with a higher frequency of female distribution towards the left, indicating that the mean person location is less than zero or below average level of health literacy (-0.337 for total items and -0.336 for 11 items) and a higher frequency of male distribution towards the right, indicating that the mean person location is more than zero or above average level of health literacy (0.175 for total items and 0.204 for 11 items). However, the wide spread of person and item locations suggests that the participants are well-targeted to items.

                    

**Table 3. Individuals' item fit statistics, and the differential item functioning for age and gender of the HQL scale.**

| Item questions | Individuals' items fit statistics of HQL scale | | | | DIF on Age: Aged 30–59 = 206, 60–75 = 100 | | | DIF on Gender: Female = 153, Male = 153 | | | |
|---|---|---|---|---|---|---|---|---|---|---|---|
| | Location | Residual | $\chi^2$ | P value | MS | F | P* | MS | F | P* | P** |
| …find information on treatments of illnesses that concern you? | -0.121 | 0.229 | 1.727 | 0.786 | 0.487 | 0.513 | 0.474 | 0.002 | 0.002 | 0.961 | 0.884 |
| …understand the leaflets that come with your medicine? | 0.22 | -1.481 | 9.107 | 0.058 | 0.712 | 0.911 | 0.341 | 4.131 | 5.313 | 0.022 | 0.025 |
| …judge the advantages and disadvantages of different treatment options? | -0.012 | 0.125 | 4.508 | 0.342 | 0.343 | 0.370 | 0.543 | 0.020 | 0.021 | 0.885 | 0.947 |
| …call an ambulance in an emergency? | -0.458 | 2.388 | 7.583 | 0.108 | 1.167 | 1.066 | 0.303 | 0.430 | 0.391 | 0.532 | 0.504 |
| …find information on how to manage mental health problems like stress or depression? | 0.114 | 0.294 | 1.588 | 0.811 | 0.200 | 0.212 | 0.645 | 0.528 | 0.559 | 0.455 | 0.522 |
| …understand why you need health screenings (such as breast exam, blood sugar test, blood pressure)? | 0.594 | -1.133 | 10.487 | 0.033 | 0.245 | 0.302 | 0.583 | 0.673 | 0.849 | 0.358 | 0.394 |
| …judge which vaccinations you may need? | 1.297 | -0.42 | 7.075 | 0.132 | 0.552 | 0.632 | 0.427 | 0.162 | 0.189 | 0.664 | 0.580 |
| …decide how you can protect yourself from illness based on advice from family and friends? | -1.692 | 0.61 | 7.461 | 0.113 | 0.033 | 0.035 | 0.852 | 2.866 | 3.013 | 0.084 | 0.073 |
| …find out about activities (such as meditation, exercise, walking, Pilates etc) that are good for your mental well-being? | -0.189 | 0.322 | 5.628 | 0.229 | 0.025 | 0.027 | 0.870 | 0.294 | 0.313 | 0.576 | 0.520 |
| …understand information in the media (such as Internet, newspaper, magazines) on how to get healthier? | 0.184 | -1.154 | 2.531 | 0.639 | 2.431 | 2.920 | 0.089 | 2.189 | 2.653 | 0.104 | 0.130 |
| …judge which everyday behaviour (such as drinking and eating habits, exercise etc.) is related to your health? | -0.086 | 0.147 | 5.432 | 0.246 | 0.208 | 0.225 | 0.636 | 3.501 | 3.881 | 0.050 | 0.042 |
| … join a sports club or exercise class if you want to? | 0.15 | 2.903 | 37.364 | 0.000 | 0.024 | 0.022 | 0.882 | 0.891 | 0.824 | 0.365 | -- |

Bonferroni adjusted p value, p = 0.004; p* is for DIF analysis; p** for DIF analysis with all items except item 12.

**Table 4. Residuals correlation matrix of the HLS-EU-Q12 scale.**

| Items* | HLQ_1 | HLQ_2 | HLQ_3 | HLQ_4 | HLQ_5 | HLQ_6 | HLQ_7 | HLQ_8 | HLQ_9 | HLQ_10 | HLQ_11 | HLQ_12 |
|---|---|---|---|---|---|---|---|---|---|---|---|---|
| HLQ_1 | -- | 0.023 | 0.114 | -0.229 | -0.148 | -0.192 | -0.336 | 0.006 | 0.008 | -0.034 | -0.065 | |
| HLQ_2 | 0.04 | -- | -0.082 | -0.177 | -0.324 | -0.009 | -0.029 | -0.18 | -0.282 | 0.124 | -0.196 | |
| HLQ_3 | 0.13 | -0.07 | -- | -0.213 | 0.1 | -0.252 | -0.294 | -0.091 | 0.079 | -0.236 | 0.005 | |
| HLQ_4 | -0.19 | -0.14 | -0.19 | -- | -0.134 | 0 | -0.053 | 0.04 | -0.191 | -0.082 | -0.248 | |
| HLQ_5 | -0.14 | -0.32 | 0.10 | -0.12 | -- | -0.183 | -0.198 | -0.096 | 0.21 | -0.245 | 0.086 | |
| HLQ_6 | -0.16 | 0.03 | -0.23 | 0.04 | -0.17 | -- | 0.094 | -0.002 | -0.224 | -0.04 | -0.165 | |
| HLQ_7 | -0.33 | -0.03 | -0.31 | -0.05 | -0.22 | 0.11 | -- | -0.121 | -0.169 | 0.058 | 0.041 | |
| HLQ_8 | 0.03 | -0.15 | -0.08 | 0.07 | -0.08 | 0.03 | -0.11 | -- | -0.037 | -0.244 | -0.181 | |
| HLQ_9 | 0.02 | -0.28 | 0.08 | -0.17 | 0.20 | -0.21 | -0.19 | -0.02 | -- | -0.302 | 0.068 | |
| HLQ_10 | -0.02 | 0.13 | -0.24 | -0.06 | -0.26 | -0.02 | 0.04 | -0.22 | -0.31 | -- | -0.144 | |
| HLQ_11 | -0.06 | -0.20 | 0.00 | -0.24 | 0.07 | -0.16 | 0.02 | -0.18 | 0.06 | -0.16 | -- | |
| HLQ_12 | -0.18 | -0.15 | -0.06 | -0.25 | 0.02 | -0.26 | 0.06 | -0.23 | -0.01 | -0.01 | 0.07 | -- |

* Residual correlation coefficients for 12 items, lower triangular part, and 11 items, upper triangular part, of the diagonal line.

## Discussion

This article aimed to assess the psychometric properties of the HLS-EU-Q12 using Rasch analysis to measure health literacy to cope with different health-related situations, including seeking health service facilities, disease prevention, and health promotion in rural Bangladesh. Overall, the assessment of the psychometric properties shows the scale performed

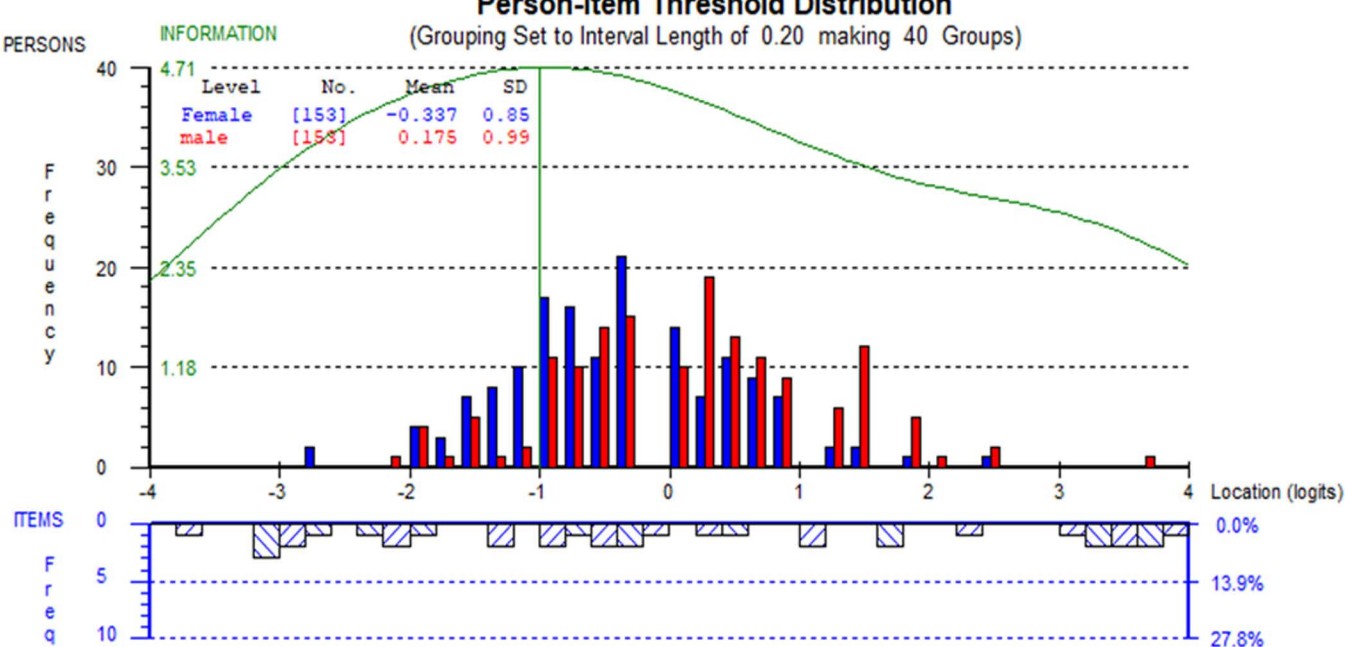

**Fig 2. Person-item location distribution of the HLS-EU-Q12 scale validated in a rural district in Bangladesh.**

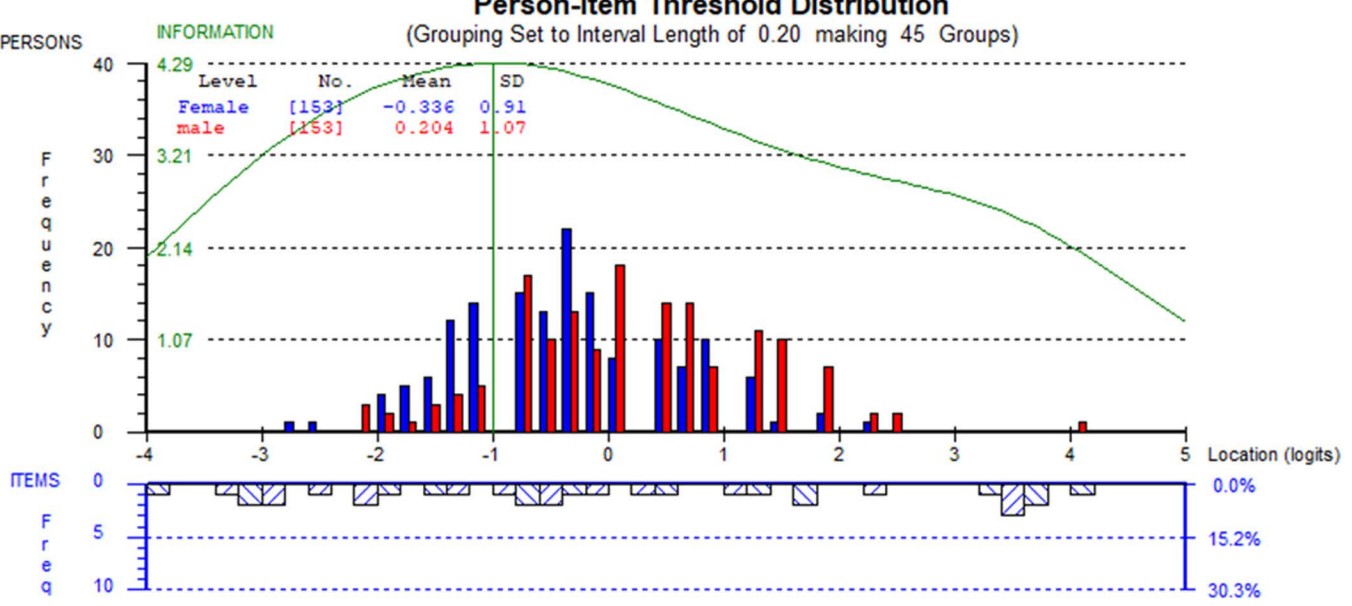

**Fig. 3. Person-item location distribution of the HLS-EU-Q12 scale validated with 11 items in a rural district in Bangladesh.**

well, with some minor weaknesses. Specifically, one correlation coefficient was 0.20 and another correlation coefficients was 0.13, and item 12's fit residual was significant. The PSI value was good, indicating that the tool's reliability is satisfactory [47]. The results indicated that all 12 items worked well with the four-point Likert-type scale. Duong et al. validated the HLS-EU-Q47 tool in six Asian countries using a large sample and reported that the questionnaire had good construct validity and goodness-of-fit of the data [62]. They concluded that the tool was a satisfactory and complete health literacy survey tool in some Asian countries. Further, in another study, Duong et al. developed the short-form HLS-SF12 questionnaire with 12 items from the original HLS-EU-Q47 tool [31]. They confirmed that the tool had good psychometric properties and high reliabilities across all six Asian countries. The current results support the previous findings and suggest that the HLS-SF12 can be a valid tool for measuring health literacy in rural areas of Bangladesh, although further investigation is warranted.

Item 12, "… join a sports club or exercise class if you want to?" had significant fit residual, which warranted a considerable variation in responding to the item. The reason for the misfit is unknown, but the question is related to personal judgment about safety, low crime rate, traffic safety, recreational facilities, and convenient places for joining sports clubs or physical activity programs, which are barriers in Bangladesh, thus can contribute to the misfit [63]. Further in-depth investigation is needed to understand this issue better. However, removing item 12 had a minor improvement in the PSI, suggesting that the item had no significant negative impact on the model. The conclusions on LD, DIF and targeting remained consistent before and after removing item 12. The residual correlation value 0.20 between items five "..find information on how to manage mental health problems like stress or depression…" and nine "…find out about activities (such as meditation, exercise, walking, Pilates etc.) that are good for your mental well-being.." can be considered high and the other correlation coefficient 0.13 for 12 items and 307 participants could be considered acceptable given that the recommended correlation coefficients 0.11 for 10 items and 0.18 for 15 items for a sample size of 300 participants [58]. Therefore, items five and nine had some redundancy information and warrant their use with caution. The HLS-Q12 showed no differential item functioning for gender or age groups. The absence of DIF supports that item parameters are stable across gender and age groups. Therefore, the current tool did not warrant modifying, rewriting, or excluding any items to eliminate possible bias toward a particular factor [64]. The person-item threshold map indicated that the items were reasonably good well-targeted to the persons. Some items were confined to a higher value of a person's location, suggesting that some items are well-responded for people with high ability. Such a response yields poor targeting, which usually decreases the reliability index of a scale [65]. The current study's reliability was not excellent, as measured by the PSI value of 0.75, due to the targeting not being at the highest level.

The strength of this study lies in its pioneering nature, being the first to assess the psychometric properties of the HLS-EU-Q12 questionnaire for use in rural areas in Bangladesh. Participants, community leaders, and a medical doctor participated actively in the research. The sample was balanced in terms of gender, eliminating any gender bias. Data were collected from face-to-face interviews, thus reducing response bias and ambiguous responses that might be caused by different levels of education, including 43% of the sample participants who had no education. The Rasch analysis in this study guided a comprehensive examination of the scale's structure.

The present study is subject to various limitations. Firstly, the results are based on the baseline data gathered for a cluster RCT, so the sample size is small. Secondly, all the measurements in this study were based on self-reports, which may have been prone to response and information bias. Thirdly, the study was carried out in only one district, which limits the ability to generalize the findings to the national level. Fourthly, although removing item 12 showed a marginally significant p-value (0.05), model fit can be recommended with caution. Item 12 is a less relevant question in a rural context in Bangladesh, where exercise or joining a sports club is not adequately available. Also, participating in sports and clubs for females is not culturally practiced well.

## Conclusion

The findings of this research demonstrate that the HLS-EU-Q12 is a potentially valid tool for measuring health literacy in rural Bangladesh. Although, the tool adheres to Rasch's assumptions of local dependency with some minor issues and unidimensionality, further study is needed to check its psychometric properties in different populations. The current

research significantly contributes to the existing body of knowledge by highlighting a crucial issue most participants in rural Bangladesh have. This study's implications are crucial for researchers, healthcare professionals, and policymakers interested in health literacy assessment and interventions in rural areas.

## Supporting Information

**Data S1  HLS EU_PLOS One.xlsx**
(XLSX)

**Supplement 1**   HLS questionnaire Bengali and English versions.pdf
(Pdf)

## Acknowledgments

We thank Professor Gavin Lambert and Elisabeth Lambert for their suggestions in developing the project. Mr. Samuel Arsovski, thank you for reviewing and proofreading the article. The author thanks the project manager Md Rafiqul Islam and the data collectors Md Abidul Islam and Helal Biswas. The author also thanks the study participants for their voluntary participation.

## Author contributions

**Conceptualization:** Fakir M Amirul Islam.

**Formal analysis:** Fakir M Amirul Islam.

**Investigation:** Fakir M Amirul Islam.

**Methodology:** Fakir M Amirul Islam.

**Writing – original draft:** Fakir M Amirul Islam.

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
