## [Decision Letter · Decision Letter 0]

3 Jan 2025

PONE-D-24-39234Study of the psychometric properties of the HLS-EU-12 questionnaire and its use in examining health literacy and the associated sociodemographic factors in rural Bangladesh.PLOS ONE

Dear Dr. Islam,

Thank you for submitting your manuscript to PLOS ONE. After careful consideration, we feel that it has merit but does not fully meet PLOS ONE’s publication criteria as it currently stands. Therefore, we invite you to submit a revised version of the manuscript that addresses the points raised during the review process.

Please address all comments from the careful reviews

We look forward to receiving your revised manuscript.

Kind regards,

Karl Bang Christensen, Ph.D.

Academic Editor

PLOS ONE

Journal Requirements:

2. Please ensure you have included the registration number for the clinical trial referenced in the manuscript.

Reviewers' comments:

Reviewer's Responses to Questions

**Comments to the Author**

1. Is the manuscript technically sound, and do the data support the conclusions?

Reviewer #1: Partly

Reviewer #2: No

2. Has the statistical analysis been performed appropriately and rigorously? 

Reviewer #1: I Don't Know

Reviewer #2: No

3. Have the authors made all data underlying the findings in their manuscript fully available?

Reviewer #1: No

Reviewer #2: No

4. Is the manuscript presented in an intelligible fashion and written in standard English?

Reviewer #1: No

Reviewer #2: Yes

5. Review Comments to the Author

Reviewer #1: Dear editor and authors,

Thank you for the opportunity to review this article. The Rasch analysis of this article presents arguments supporting that the health literacy scale under investigation has good measurement properties. However, methods and results need to be revised thoroughly, main critical points are an elusive method section and a somewhat unstructured reporting of the results. The authors have to be more explicit in describing how compliance to measurement assumptions was determined, i.e. which tests, which cut-offs and why. Also, it is not totally clear how the analysis of unidimensionality was done, and aspects of targeting are spread across the method and result sections. I would suggest to follow a clearer structure, in describing the model fit, reliability and targeting, the fit of the items and the ordering of their response options, the local item dependencies and dimensionality, as well as the differential item functioning.

The categorization of the interval scaled score may need more justification. Why are scores below 50 seen as inadequate, the smaller range from 50-65 as problematic, and the 66-84 range as sufficient? How did the authors come up with these values, it seems to be based on literature – reference is missing – a sentence on the approach could be added (is it distribution based?).

Would density curves for the scores, by gender or age group etc. not be an option with higher validity instead of barcharts for the categories?

Given that data fits the Rasch model, I would recommend considering a linear regression analysis with the interval scaled scores instead of a logistic regression where the scores have been grouped in two categories.

The English needs to be revised by a native speaker.

The authors provide good and interesting summary tables. The labels need to be adjusted and text generally controlled for typos. Correction for repeated measures have been applied whenever needed. Figure 2 is not really needed.

Here a few more comments and suggestions for correction regarding the manuscript (in order of appearance in the text).

"and the tool validated unidimensionality" -> the analysis supported unidimensionality of the scale.

Introduction:

I would delete following information that do not relate anymore to the topic of assessment of health literacy and only reports some instruments that were evaluated using the Rasch model:

Suggest to remove: "Uddin et al. [21-24] used Rasch analysis to validate the Kessler 10-item (K10) questionnaire and the World Health Organization Quality of Life (WHOQoL) 26- item (WHOQoL 26) questionnaire. They proposed a seven-item tool (K7) and 18-item (WHOQoL 18) tool, respectively in rural Bangladesh."

Page 7: 2 | MATERIALS AND ETHODS –> and METHODS

Page 8: CVD – please write it out.

Page 8: I agree, that for a 12-item tool, a sample of 300 may be sufficient. However, I am not sure if this formula is appropriate to estimate sample sizes in Rasch analysis. Why E = 0.08? Do you have a reference for using this formula in the context of Rasch analysis with polytomous items? (As I do not know, would this formula find the sample size required to conduct certain DIF analyses with higher certitude?).

Page 9: Regarding the recruitment, if the division consisting of an intervention and a control group is relevant for the scale validation, and as you mention it, I would also suggest either adding a DIF analysis for belonging to one or the other group or showing more characteristics of these groups (I am tempted to hypothesize, that the intervention group may have (acquired) higher health literacy, given their exposure to health problems and health interventions. I wonder, regarding the logistic regression, what percentage of the group with score < 50 are from the control group etc…) These two groups were not mentioned anymore later in the article.

Page 11: Suggest not to list the items in the text and rather refer to a table with the 12 items and response frequencies and percentages for each response option (maybe disaggregated by control and test group). The table can be shown in the results as part of the descriptive analysis. Meaning that the paragraph about the development of the Health Literacy tool could stop after the sentence "Finbraten et al.". Are there already published psychometrics regarding the two versions of the reported scales? these could be reported here instead.

2.8 Outcome Measures:

Scale Validation:

"fit residuals" is not a statistical test also Rasch models cannot recategorize items.

"The partial credit model recognizes that each item has its rating scale structure and is the

generalization of the dichotomous Rasch model." -> The partial credit model (PCM) is a generalization of the dichotomous Rasch model that recognizes that each item has its rating scale structure.

"Disordered thresholds usually occur when individuals find it challenging to classify between response categories, especially." [delete: for the lack of appropriate ability].

"The Rasch model requires evaluating some assumptions to ensure an instrument has Rasch properties" - > The Rasch model requires evaluating some assumptions to ensure an instrument has good measurement properties

property c) -> absence of differential item functioning

with regard to the overall fit instead of "non-significant p-value indicates that the items' hierarchical orders are stable across all underlying attribute levels."

rather write:

"A non-significant p-value indicates that the data is fitting the PCM."

The Item-person interaction statistics inform already about the targeting of the instrument, how well the items can assess the ability range of the participants. I suggest to rephrase it in a perspective of an analysis of the targeting of the instrument, where the mean difficulty and mean ability should be close to 0, (< 1 logit difference), and the SD should be within the acceptable range of +/- 2.5SD.

Page 13: "The assumption unidimensionality occurs…" -> In presence of unidimensionality all items measure one common attribute. In Rumm2030, the unidimensionality is tested with a principal component analysis of the residuals, where 1st eigenvalues < 1.8 (or 2) support the unidimensionality of the scale. (see later comment also on the result section and reporting of the dimensionality test)

"The local independence [54] can be tested by calculating the correlation coefficient

among residuals, which is expected to be unrelated.---" -> "The local independence [54] can be tested by calculating the correlation coefficient among residuals. Correlation below 0.3 indicate that two items are locally independent.

(Please see also the comment regarding result section and choice of cut-off)

"Any presence of DIF violates the requirement of independence [55]." -> Any presence… of invariance [55].

"A high PSI value indicates high reliability, indicating that the scale can separate persons along the latent trait [57]." -> A high PSI value (> 0.8?) indicates high reliability, indicating that the scale can separate persons along the latent trait [57].

"In Rasch analysis, targeting is crucial in developing or validating a scale. It refers to how well the scale captures a person's estimates based on location. The item difficulty and a person's ability are assessed by graphically and statistically comparing the distribution of the item threshold map" - > all the information regarding the targeting should be reported grouped.

I miss a section on the item fit – that could come after the model fit.

Page 14 – end of paragraph: "according to the method of the HLS-EU" -> would need a reference here.

2.9 Statistical analysis

Page 15 -"Sociodemographic factors, including sex, age, and level of education, were presented in proportion using simple descriptive statistics." -> The descriptive statistics in Table .. show the sociodemographic characteristics of the sample, including frequencies and percentages by sex, age, and level of education.

Next sentence: of these factors or of the sociodemographic factors

To some extent, I find it a pity that the authors, after the Rasch analysis and reporting good measurement properties of the scale, did not chose to apply a linear regression with the health-literacy score coded from 0-100, and instead opt for an aggregation first into 4 ordered categories (and then into two). By doing so, it was assumed that the categorical steps are homogeneous within groups. With lack of homogeneity within groups there is a risk to lose power and at worst to have an inaccurate estimation. I would want to know more on the score distributions (mean scores and variance), person parameter distribution and measurement error within these categories.

I am not sure why the authors prefer a logistic regression, after reporting a categorization into 4 (and not a multinomial regression). However, in any case, I would rather support the use of a linear model, with the 0-100 interval scaled scores. I am also fine, with the logistic regression, but would expect than a point in the discussion regarding potential limitations by doing so.

3.1 Participant's profile

is 80% of the sample unemployed or retired?

Psychometric properties of the HLS-EU-12

Page 16 - The PSI value for the 11 items, 0.756 with the same four response categories, did not show any significant improvements, indicating that removing item 12 has no prominent contribution in separating persons.

I do not agree, removing item 12, slightly increased the PSI, indicating that removing item 12 had no negative effect on the scale, to the contrary. Please rephrase accordingly.

The Unidimensionality… the method section did not describe how this analysis will be done and what cut-offs are going to be used. Please provide more information on your strategy. You applied the t-test, and lower bound being < 5% would support undimensionality, here also different approaches are possible, for example comparing the items loading positively and negatively on the first PCA component, or compare the items against items with misfit or negative correlations. How did you perform this analysis – if the PCA loadings were guiding the grouping of items, you could also report 1st eigenvalue (or 1st eig. < 1.8 also support unidimensionality).

Page 17 – What is meant by? "However, merging item 1 with 3 or 3 with 7 and for the remaining pairs when correlation coefficients were marginally higher than 0.30… ".

3.3 Health literacy and associated factors

"the female was associated", - > that does not sound appropriate. The text should be revised by a native speaker.

"one poor female housewife versus one middle – class male government employee" … Could you maybe here aggregate more and say that living in poverty, being female etc was more prevalent in the lower 5% while in the upper 5% (or also10% ) being male and employed, of younter age idk …etc was more likely to be associated with a higher health literacy score?

4 Discussion

"including a few correlation coefficients greater than 0.30, and item 12's fit residual

was significant." – including a few local item dependencies and misfit in the item assessing …

I am unsure if a PSI of 0.75 is still good – I would add the information in the method with references, usually 0.9 is excellent, 0.8 is good 0.7 is sufficient – however this may depend.

"There are, so far, no well-documented recommended critical values to conclude that indicate local dependency, and 0.30 is an arbitrary cutoff only [62-65], but a value of 0.30 has been used more often." Christensen & al recommend using a cut-off that adjusts for test length which consists in cut-off = mean residual correlation + 0.2. A cut-off of 0.3 is very relaxed, but I agree that it is commonly used.

"No DIF effect satisfies one of the underlying assumptions of Rasch models: that the estimated item parameters should be stable for different populations, such as males or females. " - > the absence of DIF supports that item parameter are stable across gender and age groups.

"The current study’s reliability was not excellent, possibly due to the targeting not being at the highest level." This final statement would contradict what is previously said regarding the PSI.

The discussion is very lengthy, and some elements of discussion would fit better in the introduction, or were already mentioned in the introduction (eg. Switzerland Taiwan example).

The authors could maybe emphasize why health literacy is important, the need for such a tool, how such a tool allows maybe to detect vulnerable groups, or specific type of info that causes difficulty. Maybe add also examples where interventions to improve the health literacy had a positive effect.

Reviewer #2: Psychometric Study Review – Feedback and Suggestions

Item Content Adjustment Post-Translation:

The psychometric study seems well-conducted; however, I am surprised that no content adjustments were made to the items during translation. The HLS-Q12 was developed in a European population, which is culturally quite different from the Bangladeshi context. Could the authors provide justification for how the item content is well-suited to the Bangladeshi population, perhaps through expert opinions on health literacy in Bangladesh?

Page 12:

The sentence:

“The HLS-EU-Q12 ranged from 19 to 44 with a mean of 30.3 (standard deviation (SD)=4.1) and a median of 30.0 from the range of 12 to 48 for 12 items with four categories.”

Is unclear in the context of Rasch model validation. Are we discussing "sum-scores," "standardized sum-scores," or latent traits here?

Page 14:

"The combined score ranged from zero (very difficult in all items) to 100 (very easy to all items), categorized into two with scores 0-50, defined as ‘limited’, and 50 and above, defined as above average health literacy. The combined score is categorized into four levels: a health literacy test score of equal or below 50% is taken as ‘inadequate’ health literacy, 51-65 as problematic, 66-83 as sufficient and greater than or equal to 84 as excellent health literacy according to the method of the HLS-EU."

How were these thresholds defined? Are these the same thresholds used in Europe, and are they relevant for a Bangladeshi population? Were these thresholds redefined based on local data?

Page 16:

"Further analysis by removing item 12 showed a good overall fit with the Rasch model, as indicated by an insignificant item-trait interaction (χ² = 60.35 df = 44, p=0.051) (data not shown).”

The statement "showed a good fit" seems an overinterpretation of a test on the verge of significance. It would be prudent to acknowledge the limitations of fit tests, which tend to retain the null hypothesis of good fit, whereas hypothesis tests are more useful for rejecting the null hypothesis. A p-value of 0.051 suggests a lack of power rather than solid evidence for model fit.

Unidimensionality Testing:

"The Unidimensionality of HLS-EU-12 was tested using principal component analysis (PCA) (5.2%, 95% CI 2.8% to 7.7%), indicating the lower bound was less than 5% (Table 2 (last row) and Fig. 3), which supports the Unidimensionality of the HLS-EU-12 item."

This sentence is unclear and does not explain that PCA was applied to residuals. What does the 5.2% represent? What reference supports the claim that this result confirms the unidimensionality of the questionnaire?

Differential Item Functioning (DIF):

Table 3 suggests potential DIF for items 10 (age) and 2, 8, 10, 11 (gender), which contradicts the statement, "However, none of the items showed significant DIFs for age and gender, indicating the scale works equally for males and females, as well as for adults and older adults."

Page 17:

"However, merging item 1 with 3 or 3 with 7 and for the remaining pairs when correlation coefficients were marginally higher than 0.30 did not improve the overall model, indicated by the chi-square value. Thus, all the items are retained."

This sentence is unclear and needs rewording.

Item 12 and Model Fit:

What should be done with item 12, which does not exhibit good fit?

Considering all these issues, can we compare the measures with those conducted in European countries?

Health Literacy Determinants Study:

The study on determinants of low health literacy seems out of place. It is not appropriate to use the same study to both validate a questionnaire (i.e., validate a measure) and explore determinants (i.e., use a measure that is assumed to already be validated). This raises concerns about using the same results for both validating the measure (known-group validity) and explaining differences in the measure.

Page 18:

The rationale for reintroducing item 12, which was previously shown to be incompatible with the measurement model, is unclear. Did this require any adaptations or statistical adjustments?

Page 20:

"HLS-SF12 can be a valid tool for measuring health literacy in rural areas of Bangladesh."

I am not entirely convinced by this statement.

DIF and Age/Gender:

"The HLS-Q12 showed no differential item functioning for gender or age groups."

This sentence does not align with the study results.

Discussion:

All discussions regarding the use of this study to explore the determinants of health literacy (good or bad) are irrelevant. The conclusion should focus on the properties of the measure itself.

Conclusion:

"The tool's adherence to Rasch assumptions of local dependency, unidimensionality, and invariance, coupled with its lack of DIF on age and sex, are significant."

This sentence is unclear and needs to be reworded.

General Conclusion:

The idea that the measure is well validated should be reworked in the conclusion and summary. These results are not impressive from a psychometric perspective and should not be overstated. At best, we have a questionnaire that is "not too bad" psychometrically.

Supplementary Documents:

It is essential for future Bangladeshi users of this questionnaire to have access to the translated version of the questionnaire.

These points highlight areas where further clarification, refinement, or additional information is necessary to enhance the psychometric rigor of the study and ensure its applicability to the Bangladeshi context.

6. PLOS authors have the option to publish the peer review history of their article (what does this mean? ). If published, this will include your full peer review and any attached files.

**Do you want your identity to be public for this peer review?** For information about this choice, including consent withdrawal, please see our Privacy Policy .

Reviewer #1: No

Reviewer #2: No

---

## [Author Response · Author response to Decision Letter 1]

19 Jan 2025

I have focused only on the psychometric properties study.

---

## [Editor Report · Decision Letter 1]

21 Jan 2025

PONE-D-24-39234R1Study of the psychometric properties of the HLS-EU-12 questionnaire in rural Bangladesh.PLOS ONE

Dear Dr. Islam,

Thank you for submitting your manuscript to PLOS ONE. After careful consideration, we feel that it has merit but does not fully meet PLOS ONE’s publication criteria as it currently stands. Therefore, we invite you to submit a revised version of the manuscript that addresses the points raised during the review process.

Thank you for resubmitting. The two reviewers did a really good job in commenting on your original submission. 

In my letter I specifically stated that 

A rebuttal letter that responds to each point raised by the academic editor and

reviewer(s). You should upload this letter as a separate file labeled 'Response to Reviewers'.

I do not think that you have managed to do this: 

- I am looking at a pdf file with 117 pages with some scattered responses at the very end.

- The file 'Response to Reviewers' contains only a single line.

please fix this. When you do so please be more careful and make sure you respond to every single question, and that you make it very clear what changes to the manuscript have been done - writing "this has been done" is typically not sufficient.

We look forward to receiving your revised manuscript.

Kind regards,

Karl Bang Christensen, Ph.D.

Academic Editor

PLOS ONE

Additional Editor Comments:

Thank you for resubmitting. The two reviewers did a really good job in commenting on your original submission.

In my letter I specifically stated that

A rebuttal letter that responds to each point raised by the academic editor and

reviewer(s). You should upload this letter as a separate file labeled 'Response to Reviewers'.

I do not think that you have managed to do this:

- I am looking at a pdf file with 117 pages with some scattered responses at the very end.

- The file 'Response to Reviewers' contains only a single line.

please fix this. When you do so please be more careful and make sure you respond to every single question, and that you make it very clear what changes to the manuscript have been done - writing "this has been done" is typically not sufficient

---

## [Author Response · Author response to Decision Letter 2]

22 Jan 2025

These are submitted as attached files for the Editor and the reviewers.

---

## [Decision Letter · Decision Letter 2]

2 Apr 2025

PONE-D-24-39234R2Study of the psychometric properties of the HLS-EU-12 questionnaire in rural Bangladesh.PLOS ONE

Dear Dr. Islam,

Thank you for submitting your manuscript to PLOS ONE. After careful consideration, we feel that it has merit but does not fully meet PLOS ONE’s publication criteria as it currently stands. Therefore, we invite you to submit a revised version of the manuscript that addresses the points raised during the review process.

 Please make sure that none of the review comments remained unaddressed.

We look forward to receiving your revised manuscript.

Kind regards,

Karl Bang Christensen, Ph.D.

Academic Editor

PLOS ONE

Journal Requirements:

Reviewers' comments:

Reviewer's Responses to Questions

**Comments to the Author**

1. If the authors have adequately addressed your comments raised in a previous round of review and you feel that this manuscript is now acceptable for publication, you may indicate that here to bypass the “Comments to the Author” section, enter your conflict of interest statement in the “Confidential to Editor” section, and submit your "Accept" recommendation.

Reviewer #1: (No Response)

2. Is the manuscript technically sound, and do the data support the conclusions?

Reviewer #1: Partly

3. Has the statistical analysis been performed appropriately and rigorously? 

Reviewer #1: No

4. Have the authors made all data underlying the findings in their manuscript fully available?

Reviewer #1: No

5. Is the manuscript presented in an intelligible fashion and written in standard English?

Reviewer #1: No

6. Review Comments to the Author

Reviewer #1: Dear author,

the revision improved the article, still the english, especially use of specific terms in the method and result section need to be more precise and unambiguous.

Not all my comments have been addressed, especially my point on local item dependencies was ignored. Based on the table with residual correlations, there are no local item dependencies in this scale (cut-off 0.3). For the local item dependencies the correlations have to be bigger than the cut-off. The authors chose 0.3, but are there correlations > 0.3. If I believe the table, there are correlations < -0.3 (of - 0.33) but negative correlations are not of interest when analysing the local item dependencies.

I suggest to the author to re-read the article more carefully, to reflect on the phrasing of methods and results. Following points have retained my attention when reading the manuscript:

General comment,

the statistical analyses per se are fine for me (except Local item dependency), however the language is not yet precise enough and the authors should invest some more time in writing everything down properly. Also, for each of the analysis done (Targeting, DIF, LID etc), the author could report the before and after deleting item 12.

Abstract: "categorization of items" – do you mean the analysis of the ordering of the response options?

"The analysis supported the scale’s unidimensionality as reported by the lower bound of a binomial 95% confidence interval (CI) of the observed proportion 5.2." – The phrasing is confusing, it sounds like 5.2 is the lower bound.

Materials and methods – the authors write that the analysis "was focused on invariance – including differential item functioning across gender and age." Why focused on invariance? and what other methods beside DIF are interested in invariance (- because of writing "including DIF").

Introduction

Please rewrite: In the study location, people have minimal knowledge of the most common chronic diseases, such as diabetes, common eye diseases, and mental health[29, 30] mental health [31].

Unidimensionality:

Please rewrite: "Compare the two estimates on a person-by-person basis using t-tests to determine the number of cases that differ significantly at the 0.05 level or the lower bound of a binomial 95% confidence interval (CI) of the observed proportion overlaps 5%. It is suggested a unidimensionality [53]."

Local Independence:

The local independence can be tested by calculating the correlation coefficients among item residuals,

which are expected to be unrelated / independent / free of any associations [54].

Differential Item Functioning:

"to test the Differential Item Functioning (DIF) of the tool." -> to test the absence of "Differential Item Functioning in the tool.

The abbreviation does not need to be repeated.

Table 2 – why not write the footnote to this table directly into the table?

Results:

Unidimensionality – this section mixes methods and results. What happens with the dimensionality after deleting item 12?

Local Dependency:

What do the authors mean with regard to the creation of the subtests that it "did not improve the overall model". It can be expected that the PSI goes down, because the dependent items inflate the reliability. In that sense, I would recommend making the subtests if there are correlations > 0.3.

However, with regard to table 4, I am wondering if the authors are also considering negative correlations. For the LID only high positive correlations are relevant. In that sense, if I am reading Table 4 correctly, the highest positive correlation is between item HLQ5 and HQL9 with r = 0.2 ? – I would rather discuss this dependency. The cut-off of 0.3 is very high.

Targeting

could the authors also add what it means, for example, if there is a higher frequency of female participants towards the left, that they have lower health literacy ? (or reverse?)

Fig 2. Typo – Perosn-item -> Person-item

Discussion-

Please correct – to what I read in the table, they are no correlations greater than 0.3. Also item 12's fit residual was significant = item 12 was misfitting.

"sufficient psychometric properties" -> had good psychometric properties?

7. PLOS authors have the option to publish the peer review history of their article (what does this mean? ). If published, this will include your full peer review and any attached files.

**Do you want your identity to be public for this peer review?** For information about this choice, including consent withdrawal, please see our Privacy Policy .

Reviewer #1: No

---

## [Editor Report · Decision Letter 3]

11 Apr 2025

Study of the psychometric properties of the HLS-EU-12 questionnaire in rural Bangladesh.

PONE-D-24-39234R3

Dear Dr. Islam,

We’re pleased to inform you that your manuscript has been judged scientifically suitable for publication and will be formally accepted for publication once it meets all outstanding technical requirements.

Kind regards,

Karl Bang Christensen, Ph.D.

Academic Editor

PLOS ONE

Additional Editor Comments (optional):

Thank you for revising

---

## [Editor Report · Acceptance letter]

PONE-D-24-39234R3

PLOS ONE

Dear Dr. Islam,

I'm pleased to inform you that your manuscript has been deemed suitable for publication in PLOS ONE. Congratulations! Your manuscript is now being handed over to our production team.

Kind regards,

on behalf of

Dr. Karl Bang Christensen

Academic Editor

PLOS ONE